# Effect of Heel-First Strike Gait on Knee and Ankle Mechanics

**DOI:** 10.3390/medicina57070657

**Published:** 2021-06-26

**Authors:** Shirin Aali, Farhad Rezazadeh, Georgian Badicu, Wilhelm Robert Grosz

**Affiliations:** 1Department of Sport Science Education, Farhangian University, Tehran 1998963341, Iran; shirin.aali1365@gmail.com; 2Department of Biomechanics and Sport Injuries, Kharazmi University of Tehran, Teheran 1998963341, Iran; rezazade.farhad@gmail.com; 3Department of Physical Education and Special Motricity, Faculty of Physical Education and Mountain Sports, Transilvania University of Brasov, 500068 Braşov, Romania; georgian.badicu@unitbv.ro

**Keywords:** biomechanics, heel strike, ankle, gait change, gastrocnemius muscle stiffness

## Abstract

*Background and Objectives*: Acquiring knowledge about the magnitude and direction of induced joint forces during modifying gait strategies is critical for proper exercise prescription. The present study aimed to evaluate whether a heel-first strike pattern during gait can affect the biomechanical characteristics of ankle and knee joints among asymptomatic people. *Materials and Methods*: In this cross-sectional study performed in the biomechanics laboratory, 13 professional healthy male athletes walked on an instrumented walkway under two walking conditions. For the normal condition, subjects were instructed to walk as they normally would. For the heel-first strike condition, subjects were instructed to walk with heel-first strike pattern and increase heel contact duration as much as possible. Then, knee and ankle joint range of motions and moments, as well as vertical ground reaction force was measured by the Kistler force plate and Vicon motion analysis system. *Results*: Knee flexion angle at the initial contact and during stance phase was significantly lower when increasing the heel strike pattern. In addition, the mean values of the knee external rotation and adductor moments during heel strike condition were lower than those in normal walking. Further, the ankle dorsiflexion range of motion (ROM) during mid-stance increased significantly during heel-first strike pattern compared to the value in normal gait pattern. *Conclusions*: The modification of gait pattern including heel-first strike pattern can reduce the mechanical load applied to the knee, while improving the extensibility of gastro-soleus muscle complex.

## 1. Introduction

Excessive joint forces play a significant role in developing musculoskeletal system impairment and pain which are the major causes of disability and form a major part of the high costs of health care in the industrialized world [1,2]. In particular, limited ankle dorsiflexion range of motion (ROM) is associated with many of the lower extremity injuries such as anterior cruciate ligament ACL, Achilles, and patella tendon injuries [3,4]. The relationship between gastro-soleus complex stiffness and many of the clinical problems of the foot and other joints of the kinetic chain are well-documented, which cause syndromes of the movement system [5,6].

Previous studies confirm the theories related to the existing mechanics of limited ankle dorsiflexion ROM and the injury of the kinetic chain joints [7,8]. The decreased ankle dorsiflexion may decrease anterior tibial translation at the ankle joint during stance phase of the gait cycle [7] and result in lowering the center of gravity during functional tasks such as walking [9]. This may compensate through mid-foot and subtalar joint pronation, knee joint flexion and valgus, which are related to chronic and acute injuries including ACL rupture, patella-femoral pain syndrome (PFPS), and knee osteoarthritis [10]. Some studies reported increased knee flexion and knee valgus during functional movements. For example, walking among people with limited ankle dorsiflexion ROM supports this theory [8,11].

Further, limited ankle dorsiflexion ROM may increase the injury risk through changing the joint forces and muscle stiffness of the lower extremities. Accordingly, the reduced ankle dorsiflexion ROM and changes in knee and hip ROM resulted in increasing ground reaction forces, which are considered as the main causes of lower extremities injuries reported in the previous studies [12,13]. Another possibility is related to the relationship between limited ankle dorsiflexion and injuries through a series of mechanical compensatory patterns instead of a unique mechanical movement pattern in one joint. According to the dynamic system theory, they are considered as multiple biomechanical degrees of freedom recruiting variability as a common output. In addition, limited dorsiflexion may demonstrate loss of a biomechanical degree of freedom (movement pattern variability) which is related to various injuries in the overall kinetic chain [14,15].

Therefore, a good understanding of how gait strategies affect moment and joint forces may improve the results of therapeutic protocols. Acquiring knowledge about the magnitude and direction of induced joint forces during modifying gait strategies is critical for prescribing proper exercise. Modification may include a subtle change in lower-extremity position such as encouraging patients to walk with heel-first strike pattern and increasing heel contact duration as much as possible. Furthermore, it is documented that the musculoskeletal system is optimized subtly to minimize demanding stresses on bones and muscles; any improper change in the movement system, such as muscle imbalance or weakness, increases the joint forces significantly [16,17]. Thus, it is important to evaluate the effectiveness of joint forces from modifying gait patterns among people with gastrocnemius muscle stiffness.

Clinically speaking, the results of the previous studies indicated that the people with gastro-soleus muscle stiffness experienced increased peak foot pronation, knee valgus, magnitude of ground reaction forces, and knee adduction moment [18,19]. Therefore, it seems that changing the gait strategy with a simple guide, such as “walking with heel-first strike pattern and increasing heel contact duration as much as possible” during walking and the extensibility of gastro-soleus muscle complex is effective, especially for the patients with limited dorsiflexion and resulting pain in other joints of the kinetic chain (lower back pain) [17]. However, during the first step of designing a therapeutic protocol, it is unclear whether modifying gait pattern with a simple guide to “heel-first strike pattern” can reduce ground reaction forces, ankle dorsiflexion during the mid-stance phase of gait, knee flexion, knee external rotation, and adduction moment among the healthy people.

Thus, the present study aimed to investigate whether a heel-first strike pattern during gait can affect the selected kinetic and kinematic parameters of ankle and knee joint among healthy subjects. Principally, it is assumed that when a subject changes their gait pattern, the vertical ground reaction force, knee flexion angle during heel contact, mean ankle dorsiflexion during mid-stance, mean knee external rotation, and adduction moment are changed to reduce joint forces and the extensibility of the gastro-soleus muscle.

## 2. Materials and Methods

### 2.1. Participants

This cross-sectional study was performed on 13 professional competitive runners with five years of experience in the national athletics team (mean age = 25.2 ± 1.2, height = 179.9 ± 1.4 cm, weight = 76.6 ± 4.7 kg) which had normal dorsiflexion ROM in both legs. The mean ankle dorsiflexion AROM in the knee extension and flexion position was 14.4 ± 0.8 and 14.5 ± 0.9 degrees, respectively. The exclusion criteria included the history of trauma or ankle surgery, bone pathology, neurological disorders, and rheumatoid arthritis, inflammatory diseases, and any conditional abnormalities affecting the research process. The individuals recruited from the available community were selected by using the findings of a preliminary study to determine the sample size based on the variance of the test parameters among five participants.

Regarding the ethical consideration, all of the participants read and signed an approved informed consent letter before data collection. The study was confirmed by the Ethical Committee of the university. Participants had the right to decline to participate and withdraw from the research after initiating the process of data collection.

The dorsiflexion ROM was measured using a universal goniometer in both knee bending for soleus length, and extended knee for gastrocnemius length positions [16,17]. At least 10 degrees of ankle dorsiflexion are needed during the stance phase of the gait cycle, which can contribute to forward body movement for normal walking [5,6]. According to previous studies, the normal ankle dorsiflexion ROM should be ranged 10–15 degrees [20,21]. Therefore, at least 12 degrees was considered as the inclusion criterion for ankle dorsiflexion AROM.

### 2.2. Gait Testing Procedures

The participants underwent gait testing during a single data collection session. All procedures were carried out at Mowafaghian gait analysis laboratory of Sharif University of technology. Standard 9 mm retroreflective markers were placed over the anatomic landmarks including the heads of the first and fifth metatarsals, the posterior aspect of the calcaneus, and the medial and lateral malleoli, lateral knee joint lines, lateral epicondyle of femur, and anterior superior iliac spine, as well as laterally on shank and thigh of the affected foot determined by anatomic definitions from the Vicon Clinical Manager [19,22]. The position data of markers and ground reaction forces were processed with Vicon dynamic model using Plug-in-Gait-Workstation software version 4.3 (Oxford, UK) to generate the kinematic data and joint moment for sequential analysis.

In the next procedure, kinematic variables during walking were collected using a six-camera, motion capture system (Motion Analysis MX40S, VICON, Oxford, UK) sampling at 120 Hz. Then, the ground reaction force data were collected from two floor-mounted Kistler force plates (30 × 50, Winterthur, Switzerland) positioned in the middle of the 6 m walkway and sampling at 1200 Hz. Accordingly, three trials with clean force platform strikes were obtained for the dominant limb. All walking trials were performed barefoot based on the participants’ self-selected and preferred walking speed.

Each participant accomplished three trials per each condition. The first is related to self-selected or preferred speed walking (normal walking). In the second condition, they were guided to increase the heel strike during walking (walking with heel-first strike pattern) based on the cue “walk with heel-first strike pattern and increase heel contact duration as much as possible”. They were given only this simple instruction without any feedback during the study.

In addition, practice trials were allowed until the participants walked comfortably and could contact the force plate with only one foot without altering their gait. Typically, three trials are performed for each situation as a practice. The stance phase was determined from the moment the heel touched the force plate until the toe was lifted off.

### 2.3. Data Analysis

The data were collected from at least three trials for each condition with clean foot strikes from each foot. Then, the moments of force were calculated by mathematical equations including inverse dynamic method and link-segment model [3,6]. To calculate the knee adduction and rotation moment, the knee moment was first calculated on the frontal and horizontal planes, and then the time series data of moment-time were normalized. The first 60% of normalized time series was considered as the stance phase. According to ISB recommendations, the positive and negative moments are considered as knee external rotation and adduction torques, respectively [3,14]. Further, joint moments were normalized to body mass (NM/kg). Furthermore, temporal parameters such as step length, walking speed, and total stance time were identified for each trial. Finally, an independent t-test was used to compare group differences after determining the normal distribution of data using the Shapiro–Wilk test and the equality of variances between groups by using Levene’s test. Statistical analysis was performed at the significance level of 0.05 and statistical power of 0.80 by SPSS software version 18.

## 3. Results

Table 1 indicates the mean and standard deviation for kinetic and kinematic variables, as well as spatiotemporal parameters of each group. Walking speed in normal walking is significantly greater than that of heel-first strike condition (*t* = 2.21, df = 12, *p* = 0.05). However, no significant difference was observed between the stance time and step length in two conditions (*t* = −0.071, df = 12, *p* = 0.94; *t* = −1.25, df = 12, *p* = 0.24, respectively). In addition, the mean values for the knee flexion-extension ROM during stance phase, knee flexion angle at initial contact, knee external rotation moment during stance phase were significantly higher in normal walking condition (*t* = 2.26, df = 12, *p* = 0.02; *t* = 3.37, df = 12, *p* = 0.005; *t* = 3.09, df = 12, *p* = 0.03, respectively). However, no significant difference was observed between the mean values for ankle dorsiflexion during stance phase in two conditions (*t* = 0.22, df = 12, *p* = 0.82). In addition, the mean ankle dorsiflexion during mid-stance was higher during walking with heel-first strike compared to normal walking (*t* = −2.37, df = 12, *p* = 0.04).

Table 2 indicates the mean and standard deviation of vertical ground-reaction force parameters. The mean value for the peak vertical force at the initial contact was significantly higher during walking with heel-first strike (*t* = −2.88, df = 12, *p* = 0.01). Further, no significant difference was reported between the mean value for peak vertical force during weight bearing in two conditions (*t* = 0.78, df = 12, *p* = 0.45). However, the mean value for peak vertical force during toe-off motion was greater during walking with heel-first strike than during the normal walking condition (*t*= −4.38, df = 12, *p* = 0.002).

## 4. Discussion

The results indicated that the ankle dorsiflexion ROM during the stance phase was greater in heel-first strike compared to the normal walking trial. Furthermore, the parameters of ground reaction force during walking with heel-first strike pattern were significantly higher than those in normal walking (Figure 1). The results could support the theory that heel-first strike during walking leads to a decrease in knee flexion angle at the initial contact and stance phase (Figure 2). In addition, two kinetic variables including the knee external rotation and adduction moment were significantly lower with increased heel-first strike trial (Figure 3). However, the spatiotemporal parameters such as step length and total stance phase time were not significantly different between trials, while walking speed in normal walking was significantly higher than walking with heel-first strike pattern condition.

Based on the results, the ground reaction forces, especially the peak impact with heel-first strike walking trial, were significantly greater than the normal walking trial (Figure 1). The findings are consistent with those of [23], which reported increased ground reaction forces in the athletes with the heel strike gait pattern. Grieve et al. [22,24] reported increased ground reaction forces due to the ankle kinematic changes, which are in line with the findings of the present study. In this study, changing the gait pattern with heel-first strike pattern resulted in changing the ground reaction force and knee joint kinematics. On the other hand, the walking speed in normal walking was significantly higher than walking with heel-first strike pattern condition (Table 1), which can be explained by increasing the heel contact duration in the walking with heel-first strike pattern condition rather than normal walking. In addition, walking with heel-first strike pattern changed the kinematics of the knee joint since a reduced inertia led to a compensatory change in the gait kinematics. However, the lack of studies in this area has created some difficulties for interpreting the results.

As displayed in Figure 2, knee flexion angle at the initial contact and the knee flexion ROM during the stance phase with heel-first strike pattern walking trial were significantly lower than those in the normal walking. Thus, the significance of the heel strike effect on ground reaction forces should be determined along with demonstrating the heel strike effect changing the knee joint kinematics in the sagittal plane. As a result, the knee and hip joints run into a more flexion position among the people with gastro-soleus stiffness compared to those in healthy people. This position puts the knee in the unlocked position leading to insufficient knee motor control, and compensatory movement patterns in the hip joint (recruiting increased hip extensor muscles and appearing to show synergistic dominance of the hamstring and gluteal muscles). Therefore, it seems that utilizing the cue “increase the heel contact duration” can change the knee flexion position during stance phase, and consequently, modify the insufficient motor control of the lower extremities.

It is believed that healthy people have a translation of tibia over the ankle joint during the stance phase of gait needed for forward propulsion. Despite the lack of significant difference of mean ankle dorsiflexion during stance phase between the trials, the mean ankle dorsiflexion ROM during mid-stance with heel-first strike pattern walking trial was significantly higher than that of normal walking (Figure 2). The finding of this pilot study may confirm that the heel-first strike walking pattern, as a functional pattern, can be effective in promoting the extensibility of the gastro-soleus muscle.

Furthermore, the values of the ground reaction force failed to decrease significantly in heel-first strike walking trial compared with those in normal walking. Thus, it is worth noting that gait strategies should be modified to achieve a special purpose, not as a set of positive effects. For example, increasing the ankle dorsiflexion ROM and decreasing the knee flexion angle for the individuals with gastro-soleus tightness are more essential than increasing ground reaction forces during walking with heel-first strike pattern since limited ankle dorsiflexion and increased knee flexion are considered as the risk factors for musculoskeletal injuries, such as patellofemoral pain syndrome, osteoarthritis, as well as synergistic dominance of hamstring muscles [10,15,16].

The results indicated that the mean external rotation and adduction moment of the knee in the heel-first strike pattern gait trial decreased significantly compared with those in a normal gait (Figure 3). In the present study, the knee external rotation and adduction moment were measured to identify and establish a movement pattern for the purpose of decreasing mechanical loading on the knee through the guidance to heel-first strike pattern since the improper loading of the knee is regarded as a risk factor for the knee osteoarthritis among the people with gastro-soleus tightness based on the literature [10,20]. In particular, medial knee compartment loading is considered as a clinical indication of the knee injuries and the knee external rotation and adduction moments are mentioned as an indirect measure of the medial knee compartment loading during functional tasks such as walking in most of the recent studies [20,21]. Therefore, early detection of the risk factors involved in knee osteoarthritis, as well as identifying the effective movement pattern for reducing the knee external rotation and adduction moment values may be more successful with the effectiveness of exercise interventions and preventing structural changes in the knee joint. Based on the results of the present study, walking with a heel-first strike pattern can reduce the loading forces of the knee joint. However, the findings cannot be generalized to those with gastro-soleus tightness. Therefore, further research is needed to ensure the effectiveness of heel-first strike gait pattern on reducing mechanical loading on the kinetic chain in people with gastro-soleus tightness.

## 5. Conclusions

Based on the results, heel-first strike gait pattern in healthy athletes could make kinematic changes in the knee and ankle joints in all three movement planes. For example, knee flexion angle at the initial contact and the knee flexion ROM during the stance phase decreased with heel-first strike pattern walking. Additionally, some kinetic changes including the mean values of the knee external rotation and adductor moments during heel strike condition were lower than those in normal walking. In addition, heel-first strike gait pattern with increased ankle dorsiflexion at the heel contact led to the extensibility of gastro-soleus muscle complex.

The main limitation is that the participants had no limited dorsiflexion ROM. Thus, the people with gastro-soleus tightness failed to respond to heel-first strike gait pattern like normal people. The present pilot study aimed to evaluate the concept of heel-first strike gait pattern for decreasing knee flexion, increasing ankle dorsiflexion, minimizing knee external rotation and adduction moment during gait, and applying the pattern in people with gastro-soleus tightness. By confirming the heel-first strike gait pattern effect on kinematic changes in this study, another study can be conducted on applying heel strike gait in the athletes with gastro-soleus tightness.

## Figures and Tables

**Figure 1 medicina-57-00657-f001:**
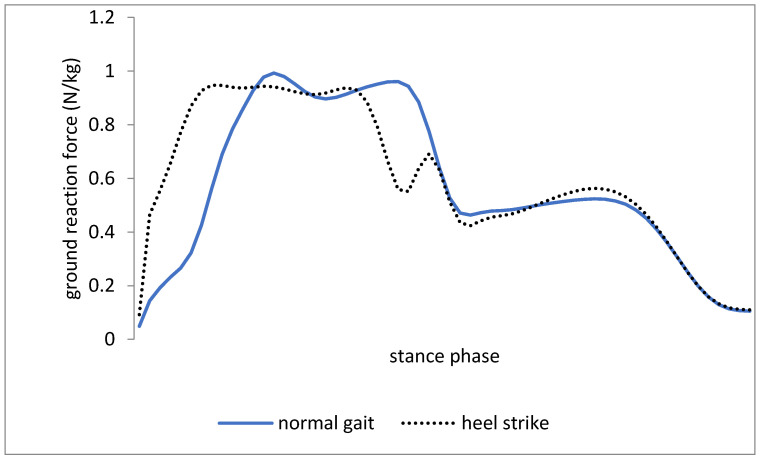
Ground reaction force during increased heel strike gait and normal gait during stance phase.

**Figure 2 medicina-57-00657-f002:**
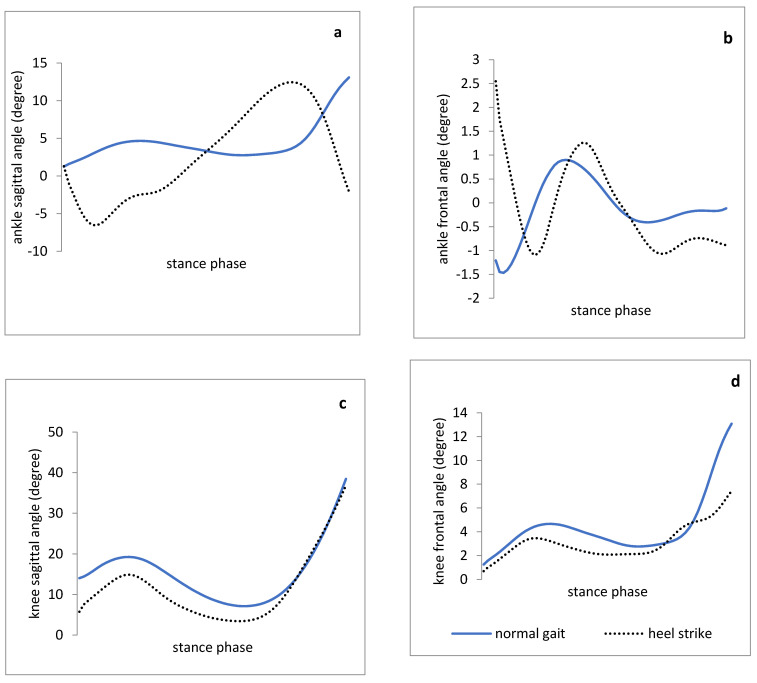
Joint angle patterns (degree) of healthy athletes with increased heel strike gait and normal gait during stance phase. (**a**,**b**): ankle joint angles in the sagittal and frontal plane, respectively. (**c**,**d**): knee joint angles in the sagittal and frontal planes, respectively.

**Figure 3 medicina-57-00657-f003:**
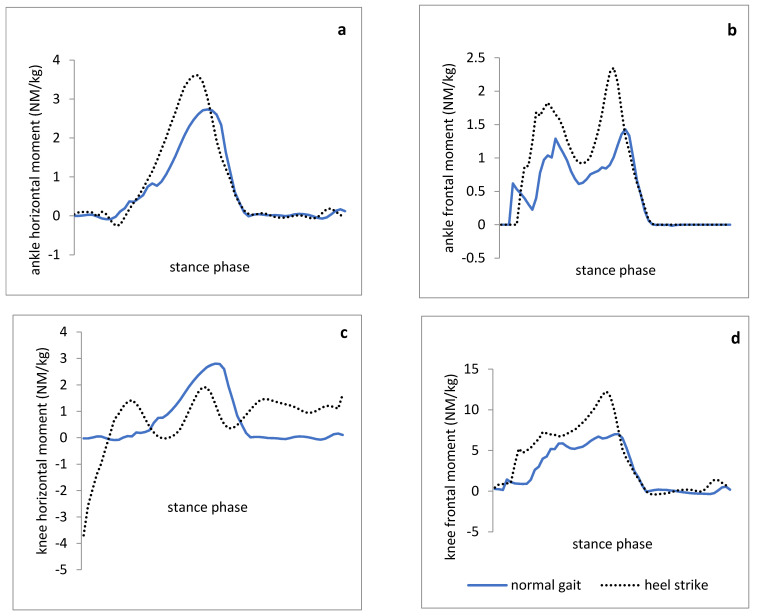
Joint moment pattern (N.M/Kg) of healthy athletes with increased heel strike gait and normal gait during stance phase. (**a**,**b**): ankle moment in the horizontal and frontal planes, respectively. (**c**,**d**): knee joint moment in the horizontal and frontal planes, respectively.

**Table 1 medicina-57-00657-t001:** The mean (standard deviation) for kinetic, kinematic, and spatiotemporal parameters.

Variable	Walking with Increased Heel Strike	Normal Walking	*p* Value	Cohen’s D
Knee flexion-extension ROM during stance phase (degrees)	23.6 (7.3)	32.8 (12.7)	0.02 *	−1.34
Knee flexion angle at initial contact (degrees)	7.1 (3.7)	14.02 (6.4)	0.005 *	−1.32
Mean ankle dorsiflexion during stance phase (degrees)	7.9 (6.5)	5.6 (4.4)	0.82	0.41
Mean ankle dorsiflexion during mid-stance (degrees)	13.84 (2)	11.47 (1.5)	0.04 *	1.18
Mean knee external rotation moment during stance phase (NM/Kg)	0.18 (0.03)	0.23 (0.05)	0.03 *	−1.21
Mean knee adduction moment during stance phase (NM/kg)	4.01 (2.7)	7.6 (3.3)	0.01 *	−1.19
Total stance phase time (second)	0.79 (0.12)	0.79 (0.14)	0.94	0
Walking speed (m/s)	1.05 (0.02)	1.16 (0.16)	0.05	−0.96
Step length (m)	1.32 (0.11)	1.26 (0.08)	0.24	0.62

ROM: range of motion; * indicates statistical significance.

**Table 2 medicina-57-00657-t002:** The mean (standard deviation) of ground reaction force parameters (data expressed as percentage of stance phase time).

Vertical Ground Reaction Force (N/kg)	Walking with Increased Heel Strike	Percentage of Stance Phase	Normal Walking	Percentage of Stance Phase	*p* Value	Cohen’s D
The peak vertical force at the initial contact	0.53 (0.07)	0.8	0.35 (0.19)	1.2	0.01 *	−1.25
Peak vertical force during weight bearing	0.98 (0.06)	6	1.02 (0.16)	8	0.45	0.33
Peak vertical force during toe-off motion	1.09 (0.03)	20	1.05 (0.04)	24	0.002 *	−1.13

* indicates statistical significance.

## Data Availability

The data presented in this study are available on request from the author Farhad Rezazadeh.

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
