# Peer review of "Effect of Heel-First Strike Gait on Knee and Ankle Mechanics"

_medicina, 2021, doi:10.3390/medicina57070657_

Round 1

Reviewer 1 Report

Paper title: Effect of Heel-first strike pattern on the Knee etc Authors: Aali et al. 
Dear Authors,
1. your paper is very well written;
2. your results should be published to the benefit of athletes;
3. the paper misses a few details: unclear sentences, non-explained acronyms, non-referred instruments, . . . I list a number of them below;
4. I recommend to display a visual about your knowledge domain, some of your readers will be athletes and they might be lost because of anatomical terms like gastrocnemius, etc. An example for such idea for layman readers, taken from Google images is not possible to copy in here;
5. I recommend shortening the title of the paper. Shorter titles are more often referred to (according to research from publisher Elsevier’s Mendeley team). A shorter title could be: Effect of Heel-first strike gait on Knee and Ankle mechanics;
6. I recommend including details of used instruments and apparatus (in the references list, for instance). Why? Because you are seemingly unclear about these machines. Examples:
   1. Line 25, reference to the company of Kistler and Vicon apparatus is missing;
   2. Line 129, is another type of machine mentioned than at Line 25? Kistler is removed;
   3. Line 124, is seemingly a riddle to me: anatomic definitions are from Vicon’s Manager;
7. typos, listed below:
   4. Line 20, please insert which lab. The Kharazmi laboratory?
   5. Line 25, reference to the company of Kistler apparatus is missing;
   6. Line 30, the acronym ‘ROM’ should be explained once in the beginning;
   7. Line 35, are keywords missing? gait change, gastrocnemius muscle stiffness;
   8. Line 41, the acronym ‘ACL’ should be explained once in the beginning;
   9. Line 62, “there” should be “they” I think;
 10. Line 63, the sentence is incorrect it seems: “degrees of freedom recruiting variability as a common output”;
 11. Lines 72 – 75, hold a very long sentence. Please help the reader here. For instance by putting between commas: like muscle imbalance or weakness;
12. Line 91, I recommend talking about subjects, stead of ‘persons’. If you agree, then ‘his gait’ should read ‘its gait’;
13. Line 114, misses a dot at the end of the sentence. Are there more missing?
14. Line 117 – 118, a blank line is missing above 2.2. Gait testing procedures;
15. Line 139, ‘only this simple instruction’ seems missing to me;
16. Lines 228 – 229, miss a plural verb: a, b show and c, d show;
17. Line 241, pal nes is wrong;
18. Line 256, what is the theory “Which heel-strike”?
19. Lines 289 – 293, hold one single sentence. This sentence is incomprehensible. Please split up;
20. Lines 305 – 307, seem to repeat from the Lines 257 – 258. Could you split up the discussion in – for instance biomechanical and spatiotemporal –, to show that the discussion is not repetitive?
 21. Lines 324 – 331, about limitations on the research could be appropriately put in next Chapter

Conclusions;

22. Line 350, “Thank numerous…” misses ‘to’, or you could say “Thanks to”;

Author Response

Dear Reviewer, 

Please see the answers.

Response to reviewer’s comments:

Effect of Heel-first strike gait pattern on the Knee and Ankle
Biomechanics of Asymptomatic Individuals: A fine modification in exercise

 Shirin Aali, Farhad Rezazade, Georgian Badicu and, Wilhelm Robert Grosz *

Journal: Medicina

Sport science section

Reviewer #1

We thank Reviewer #1 for their valuable comments and suggestions, which helped us to substantively improve the quality of the manuscript.

1. your paper is very well written;

2.your results should be published to the benefit of athletes;

We thank Reviewer #1 again for their positive, valuable and encouraging comments and suggestions.

3. The paper misses a few details: unclear sentences, non-explained acronyms, non-referred instruments, I list a number of them below;

4. I recommend to display a visual about your knowledge domain, some of your readers will be athletes and they might be lost because of anatomical terms like gastrocnemius, etc. An example for such idea for layman readers, taken from Google images is not possible to copy in here;

We completely agree with honorable reviewer idea, but there is tow problem here: 1- there is a limitation for number of picture, table and graphs for journal. 2- The second is related to copy right problems in using images. So this changed remained.

5. I recommend shortening the title of the paper. Shorter titles are more often referred to (according to research from publisher Elsevier’s Mendeley team). A shorter title could be: Effect of Heel-first strike gait on Knee and Ankle mechanics;

Thank you for your practical suggestion, the title reads now:

Effect of Heel-first strike gait on Knee and Ankle mechanics

6. I recommend including details of used instruments and apparatus (in the references list, for instance). Why? Because you are seemingly unclear about these machines. Examples:
   1. Line 25, reference to the company of Kistler and Vicon apparatus is missing;
   2. Line 129, is another type of machine mentioned than at Line 25? Kistler is removed;
   3. Line 124, is seemingly a riddle to me: anatomic definitions are from Vicon’s Manager;

Thank you again for helping us to correct this issue. We carefully checked and correct them as below:

1-     We referred to company in line 129: the text reads now: In the next procedure, kinematic variables during walking were collected using a six camera, motion capture system (Motion Analysis MX40S, VICON, USA) sampling at 120 Hz. Then, the ground reaction force data were collected from two floor-mounted Kistler force palate (30*50, Switzerland)

2-     Thank you for pointing us attention to this issue, the text reads now: In the next procedure, kinematic variables during walking were collected using a six camera, motion capture system (Motion Analysis MX40S, VICON, USA) sampling at 120 Hz. Then, the ground reaction force data were collected from two floor-mounted Kistler force palate (30*50, Switzerland).

3-     by anatomic definitions from the Vicon Clinical Manager

7. typos, listed below:
   4. Line 20, please insert which lab. The Kharazmi laboratory?
   5. Line 25, reference to the company of Kistler apparatus is missing;
   6. Line 30, the acronym ‘ROM’ should be explained once in the beginning;
   7. Line 35, are keywords missing? gait change, gastrocnemius muscle stiffness;
   8. Line 41, the acronym ‘ACL’ should be explained once in the beginning;
   9. Line 62, “there” should be “they” I think;
 10. Line 63, the sentence is incorrect it seems: “degrees of freedom recruiting variability as a common output”;
 11. Lines 72 – 75, hold a very long sentence. Please help the reader here. For instance by putting between commas: like muscle imbalance or weakness;
12. Line 91, I recommend talking about subjects, stead of ‘persons’. If you agree, then ‘his gait’ should read ‘its gait’;
13. Line 114, misses a dot at the end of the sentence. Are there more missing?
14. Line 117 – 118, a blank line is missing above 2.2. Gait testing procedures;
15. Line 139, ‘only this simple instruction’ seems missing to me;
16. Lines 228 – 229, miss a plural verb: a, b show and c, d show;
17. Line 241, pal nes is wrong;
18. Line 256, what is the theory “Which heel-strike”?
19. Lines 289 – 293, hold one single sentence. This sentence is incomprehensible. Please split up;
20. Lines 305 – 307, seem to repeat from the Lines 257 – 258. Could you split up the discussion in – for instance biomechanical and spatiotemporal –, to show that the discussion is not repetitive?
 21. Lines 324 – 331, about limitations on the research could be appropriately put in next Chapter Conclusions;
 22. Line 350, “Thank numerous…” misses ‘to’, or you could say “Thanks to”;

7. Thank you again for helping us to correct this issues. We carefully checked and correct as following:

4.the lab name inserted in the text to line 117 and 118 as below:  All procedures was carried out at Mowafaghian gait analysis laboratory of Sharif University of technology

5. The company name is inserted to the text.

6. the ROM explanation inserted to the text in line 28, the text reads now:

…. the ankle dorsiflexion (range of motion) ROM during mid-stance increased significantly…..

7. thank you, the key words added to this section as below:

Keywords: Biomechanics; Heel Strike; Ankle; Gait change; Gastrocnemius Muscle Stiffness

8. Thank you. The explanation added to the line 48 as below:

….injuries including (anterior cruciate ligament) ACL rupture….

9. Thank you. The text reads now:

…., they are considered as multiple biomechanical….

10. the sentence corrected as reviewer comment:

The text reads now:

According to the dynamic system theory, they are considered as multiple biomechanical degrees of freedom recruiting variability as a common output.

11. the text reads now:

Furthermore, it is documented that the musculoskeletal system is optimized subtly to minimize demanding stresses on bones and muscles; because any improper change in the movement system like muscle imbalance or weakness, increases the joint forces significantly.

12. Thank you. we corrected this according to honorable reviewer comment: the text reads now:

….parameters of ankle and knee joint among healthy subjects. Principally, it is assumed that when a subject changes its gait pattern, the vertical ground reaction force, knee flexion…..

13. Thank you for your detailed review of our paper. We checked entire manuscript and fix it. 

14. We added the blank line above next title.

15. Thank you. This phrase refers to prior sentence. So we corrected it as below:

They were given only this simple instruction without any feedback during the study.

16. We corrected this accordingly.

 17. thank you again. We corrected this to “planes”.

18.  the theory  is : “heel-first strike during walking leads to a decrease in knee flexion angle at the initial contact and stance phase”  so the text reads now:

The results could support the theory that heel-first strike during walking leads to a decrease in knee flexion angle at the initial contact and stance phase.

19. Thank you. the text reads now:  

Despite the lack of significant difference of mean ankle dorsiflexion during stance phase between the trials, the mean ankle dorsiflexion ROM during mid-stance with heel-first strike pattern walking trial was significantly higher than that of normal walking (Fig. 2).

20. As you know main results briefly described in the first paragraph of the discussion and explained in details in the following paragraphs. And since some variables discussed together, separating the topics will lead to a repetition of some sentences in the discussion. So with respect to honorable reviewer suggestion this part remained unchanged but the repeated sentence corrected as bellow:   

In addition, tow kinetic variables including the knee external rotation and adduction moment were significantly lower with increased heel-first strike trial

21. We put the limitation under the conclusion.

22. The text reads now: Thank to numerous individuals participated in this study.

We thank Reviewer #1 again for their positive, valuable and encouraging comments and suggestions.

Thank you a lot!

Reviewer 2 Report

Thank you very much for the opportunity to review this manuscript.

The research topic is very interesting. However, there were a few places that authors should respond and fix as indicated below. There are some mistakes of English grammar so that the manuscript should be checked by a native English speaker.

Line 22: “natural condition” should be fixed to “normal condition” because “normal condition” was used in other places.

Line 25: “force plate” should be added after “Kistler” to explain the instrument.

Line 98: To understand the subject characteristics, please add the information of their sports.

Line 100: “14.4 ± 0.84” should be fixed to “14.4 ± 0.8” because of significant digits.

Line 112: “shortness” should be “lengthened.”

Table 1 and Table 2: Significant digits should be unified.

Figure 1: Put the unit of y-axis and explanation of stance phase (1~59).

Figure 2: The graph (b) was not clear. Please fix it. X-axis is not clear too.

Put the unit of graph (a) and (b).

Figure 3: The graph (b) was not clear. Please fix it.

Line 332: Conclusion should be the summary of the study results (what the authors found). Please rephrase.

Author Response

Dear Reviewer, 

Please see the answers.

Reviewer 2#

The research topic is very interesting. However, there were a few places that authors should respond and fix as indicated below. There are some mistakes of English grammar so that the manuscript should be checked by a native English speaker.

Line 22: “natural condition” should be fixed to “normal condition” because “normal condition” was used in other places.

Line 25: “force plate” should be added after “Kistler” to explain the instrument.

Line 98: To understand the subject characteristics, please add the information of their sports.

Line 100: “14.4 ± 0.84” should be fixed to “14.4 ± 0.8” because of significant digits.

Line 112: “shortness” should be “lengthened.”

Table 1 and Table 2: Significant digits should be unified.

Figure 1: Put the unit of y-axis and explanation of stance phase (1~59).

Figure 2: The graph (b) was not clear. Please fix it. X-axis is not clear too.

Put the unit of graph (a) and (b).

Figure 3: The graph (b) was not clear. Please fix it.

Line 332: Conclusion should be the summary of the study results (what the authors found). Please rephrase.

We thank Reviewer #2 for their valuable comments and suggestions, which helped us to substantively improve the quality of the manuscript.

We corrected all sentences according to honorable reviewer comments as below:

Line 22: corrected to “normal condition”

Line 25: we added “force plate” after kistler

Line98:the text reads now: This cross-sectional study was performed on 13 professional competitive runners with five years of experience in the national athletics team

Line100: we fixed it to “14.4 ± 0.8”

Line112: the text reads now:

… both knee bending for soleus length, and extended knee for gastrocnemius length positions….

Table 1 and table2:

We reported significant digits Up to two decimal places but in one case in the table 1(Knee flexion angle at initial contact (degrees)) and one case in the table 2 (Peak vertical force during toe off) we had to report significant digit up to three decimal places because of zero. So Therefore, with respect to the opinion of the referee this part remained unchanged.

Figure 1: we add some changes to the graph and resolved this problem.

Figure 2: the problem resolved. The units has been added to the graph.

Figure 3: the problem is fixed.

Line 332: the conclusion rephrase as much as possible: the text reads now:

 Based on the results, heel-first strike gait pattern in the healthy athletes could make kinematic changes in the knee and ankle joints in all three-movement planes. For example Knee flexion angle at the initial contact and the knee flexion ROM during the stance phase decreased with heel-first strike pattern walking. Also, some kinetic changes including the mean values of the knee external rotation and adductor moments during heel strike condition were lower than those in normal walking. In addition, heel-first strike gait pattern with increased ankle dorsiflexion at the heel contact led to the extensibility of gastro-soleus muscle complex.

The main limitation is that the participants had no limited dorsiflexion ROM. Thus, the people with gastro-soleus tightness failed to respond to heel-first strike gait pattern like normal people. The present pilot study aimed to evaluate the concept of heel-first strike gait pattern for decreasing knee flexion, increasing ankle dorsiflexion, minimizing knee external rotation and adduction moment during gait before clinical, and applying the pattern in people with gastro-soleus tightness. By confirming the heel-first strike gait pattern effect on kinematic changes in this study, another research can be conducted on applying heel strike gait in the athletes with gastro-soleus tightness.

Thank you a lot!

Round 2

Reviewer 2 Report

The authors have thoroughly considered the suggestions and criticisms from the reviewers and revised the manuscript accordingly.